# Doxorubicin induces prolonged DNA damage signal in cells overexpressing DEK isoform-2

Emrah Özçelik[1], Ahmet Kalaycı[1], Büşra Çelik[1], Açelya Avcı[1], Hasan Akyol ⬢[1], İrfan Baki Kılıç[1], Türkan Güzel[1], Metin Çetin[1], Merve Tuzlakoğlu Öztürk[1], Zihni Onur Çalışkaner[1]¤, Melike Tombaz[2], Dilan Yoleri[1], Özlen Konu[2], Ayten Kandilci ⬢[1]*

1 Department of Molecular Biology and Genetics, Gebze Technical University, Gebze, Kocaeli, Turkey,
2 Department of Molecular Biology and Genetics, Bilkent University, Ankara, Turkey

¤ Current address: Department of Molecular Biology and Genetics, Biruni University, Istanbul, Turkey
* akandilci@gtu.edu.tr

**Citation:** Özçelik E, Kalaycı A, Çelik B, Avcı A, Akyol H, Kılıç İB, et al. (2022) Doxorubicin induces prolonged DNA damage signal in cells overexpressing DEK isoform-2. PLoS ONE 17(10): e0275476. https://doi.org/10.1371/journal.pone.0275476

**Data Availability Statement:** All relevant data are within the paper and its Supporting Information files.

## Abstract

DEK has a short isoform (DEK isoform-2; DEK2) that lacks amino acid residues between 49–82. The full-length DEK (DEK isoform-1; DEK1) is ubiquitously expressed and plays a role in different cellular processes but whether DEK2 is involved in these processes remains elusive. We stably overexpressed DEK2 in human bone marrow stromal cell line HS-27A, in which endogenous DEKs were intact or suppressed via short hairpin RNA (sh-RNA). We have found that contrary to ectopic DEK1, DEK2 locates in the nucleus and nucleolus, causes persistent γH2AX signal upon doxorubicin treatment, and couldn't functionally compensate for the loss of DEK1. In addition, DEK2 overexpressing cells were more sensitive to doxorubicin than DEK1-cells. Expressions of *DEK1* and *DEK2* in cell lines and primary tumors exhibit tissue specificity. *DEK1* is upregulated in cancers of the colon, liver, and lung compared to normal tissues while both *DEK1* and *DEK2* are downregulated in subsets of kidney, prostate, and thyroid carcinomas. Interestingly, only *DEK2* was downregulated in a subset of breast tumors suggesting that *DEK2* can be modulated differently than *DEK1* in specific cancers. In summary, our findings show distinct expression patterns and subcellular location and suggest non-overlapping functions between the two DEK isoforms.

## Introduction

DNA damage and repair is one of the challenges that living cells face at any moment and the long-term survival of a healthy cell depends on the successful overcoming of that challenge. DEK is a versatile nuclear protein involved in many biological processes including chromatin remodeling, regulation of transcription, splicing, and DNA repair (reviewed in [1–3]). *DEK* gene encodes two isoforms generated via alternative splicing resulting in either full-length DEK (DEK1) or the spliced form (DEK2), containing 375 and 341 amino acids, respectively. Ubiquitously-expressed DEK1 was subjected to intense research since its discovery in 1992 as a part of abnormal DEK-CAN fusion [4], and it was shown that many solid tumors

**Funding:** Prof. Dr. Ayten Kandilci Grant number: 118Z765 Funding: Scientific and Technological Research Council of Turkey (TÜBİTAK) https:// www.tubitak.gov.tr/ The funders had no role in study design, data collection and analysis, decision to publish, or preparation of the manuscript. Mr. Emrah Özçelik Grant number: 216Z006 Funding: Scientific and Technological Research Council of Turkey (TÜBİTAK) https://www.tubitak.gov.tr/ The funders had no role in study design, data collection and analysis, decision to publish, or preparation of the manuscript.

**Competing interests:** The authors have declared that no competing interests exist.

overexpress DEK1 [5–8]. Consistently, increased level of DEK1 in cancerous epithelial cells due to gene amplification or transcriptional activation was associated with higher proliferation capacity, whereas lack of DEK1 expression was related to decreased cell survival and lower tumor incidence in mouse models [9–11]. Depending on the cell type and cellular context, DEK1 exists in the nucleus and/or in the cytoplasmic granules and contributes to different cellular processes. About 90% of DEK1 is bound to chromatin, and phosphorylation or acetylation reduces its binding [12–14]. DEK1 also acts as a chemotactic factor for hematopoietic cells when secreted from the activated macrophages and neutrophils [15–17]. Despite all these data covering DEK1, the function of DEK2 remains elusive. A recent study indicated that DEK1 stimulates migration and proliferation more efficiently than DEK2 when overexpressed in the hepatocellular carcinoma cells, suggesting a functional difference between the two isoforms [18]. Since there is not an antibody specific to the endogenous DEK2, currently it is not possible to analyze/discriminate endogenous DEK1 and DEK2 proteins using Western blot analysis or immunofluorescence staining. Therefore, here we elaborated the expression level of the two isoforms of DEK in selected cell lines and primary cancer samples using RT-qPCR and bioinformatics analysis of public datasets. We also compared the cellular location of ectopically expressed epitope-tagged DEK isoforms, and analyzed their effect on the proliferation, doxorubicin sensitivity, and DNA repair capacity in HS-27A cells, which relatively expresses higher endogenous *DEK2*.

## Materials and method

### Cloning

*DEK1* cDNA that was obtained from HS-27A cells was tagged by Flag or Myc epitope at the 5' using "Flag-forward", "Myc-forward", and the common "Reverse-1" primers, and the epitope-tagged cDNA was cloned into the pGEM-T Easy Vector (Promega, Madison, WI, USA, A1360). *DEK2* cDNA was generated from this pGEM-T plasmid containing the epitope-tagged *DEK1* cDNA insert using site-directed mutagenesis and the primer sets of "Site-direct DEK2-forward" and "Site-direct DEK2-reverse". Then, pGEM-T plasmids containing either *DEK1* or *DEK2* cDNA were cut with the EcoR1 enzyme (New England Biolabs, R3101S) and sub-cloned into the EcoR1 site of the MSCV-IRES-GFP (MIG) retroviral vector.

*DEK1GFP* or *DEK2GFP* fusions (pDEK1GFP or pDEK2GFP) were generated by cloning the HA epitope-tagged *DEK1* or *DEK2* cDNA into the XhoI/EcoRI cloning site of the pEGFP-N2 (Clontech, Mountain View, California, USA, 6081–1) vector using the "Xho1-HA-Forward" and "EcoR1-Reverse primers". The integrity of the inserts in all vectors was confirmed by sequencing. All primers were listed in Table 1.

### Generation of cell lines

Cells were cultured in Dulbecco's Modified Eagle Medium (DMEM) (for 293T, HeLa) or RPMI-1640 medium (for HS-27A) containing 10% fetal bovine serum and 1% penicillin/streptomycin (Gibco, Thermo Fisher Scientific, Waltham, MA, USA, 15140122) at 37°C with 5% $CO_2$ atmosphere.

Stable overexpression of epitope-tagged DEK isoforms (HS-27A-DEK1-GFP, HS-27A-DEK2-GFP) or the knockdown by sh-RNA (Origene, Rockville, MD, USA, TL313507) of endogenous DEK isoforms in HS-27A cells (HS-27A-shDEK) performed as described before using the corresponding VSVg pseudotyped MIG retrovirus or sh-RNA lentivirus [19]. HS-27A-DEK1-GFP, HS-27A-DEK2-GFP, and the control cells transduced with an empty MIG (HS-27A- GFP) were sorted by BD FACSJazz™ (BD Bioscience, Franklin Lakes, New Jersey, U.S.), and over 90% GFP-positive cells were obtained (S1 Fig). HS-27A-sh-negative and HS-

**Table 1. List of used primers.**

| Primer | Primer Sequences (5'-3') |
|---|---|
| Flag-Forward | ATGGACTACAAAGACGATGACGACAAGTCCGCCTCGGCCCCTG |
| Myc-Forward | ATGGAACAAAAACTCATCTCAGAAGAGGATCTGTCCGCCTCGGCCCCTGCTG |
| Reverse-1 | TCAAGAAATTAGCTCTTTTACAG |
| Site-Direct DEK2-Forward | GGAGGAAAAAGGAAAGGGGCAGAAACTTTGTGA |
| Site-Direct DEK2-Reverse | GCCCCTTTCCTTTTTCCTCCTCCTCCTCCTCCT |
| Xho1-HA-Forward | TAAGCACTCGAGATGTACCCATACGATGTTCCAGATTACGCTTCCGCCTCGGCCCCTGCTG |
| EcoR1-Reverse | TGCTAAGAATTCTGCTTATCAAGAAATTAGCTCTTTTACAG |
| DEK1-Forward | GTCTCATCGTGGAAGGCAAG |
| Reverse-2 | GTGCCTGGCCTGTTGTAAAG |
| DEK2-Forward | TCCGAGAAAGAACCCGAAAT |
| Reverse-3 | CTGCCCCTTTCCTTTTTCCT |
| GAPDH-Forward | GAAGGTGAAGGTCGGAGTC |
| GAPDH- Reverse | GAAGATGGTGATGGGATTTC |
| DEK2-Forward-2 | ATGTCCGCCTCGGCCCCTG |
| ACTB-Forward | CAGCCATGTACGTTGCTATCCAGG |
| ACTB-Reverse | AGGTCCAGACGCAGGATGGCATG |

27A-shDEK cells were selected with 1.25μg/ml puromycin for 10 days. After selection and propagation, the HS-27A-shDEK cells were re-infected with MIG-FlagDEK2 or MIG-Flag-DEK1 retrovirus to obtain overexpression.

Transient transfections of HeLa cells with pDEK1GFP or pDEK2GFP expression vectors were performed using the Profection Mammalian Transfection System (Promega, WI, USA, E1200) according to the manufacturer's instruction and the cells were used in the corresponding experiments 24 or 48 hr after the transfection.

## RT-PCR analyses

After the total RNA extraction with TriPure Isolation Reagent (Roche, Basel, Switzerland, 11667157001) and synthesis of cDNA using 500ng of total RNA using a High-Capacity cDNA Reverse Transcriptase Kit (Applied Biosystems, Thermo Fisher Scientific, 4368815), the semi-quantitative RT-PCR reactions were performed using GoTaq Green Master Mix Kit (Promega, M7122) and the primer pairs of DEK1-forward and Reverse-2; DEK2-Forward and Reverse-3; GAPDH-forward and GAPDH-reverse. MIG plasmids containing the cDNA *of DEK* isoforms were used as a positive control in the PCR reactions. RT-qPCR reactions of the cell lines were performed by using the Power SYBR$^{TM}$ Green PCR Master Mix Kit (Applied Biosystems, 4368706) and a StepOnePlus Real-Time PCR System (Applied Biosystems, 4376600) and the following primers: DEK1-forward and Reverse-2; DEK2-forward-2 and Reverse-3; GAPDH-forward and GAPDH-reverse. GAPDH (glyceraldehyde-3-phosphate dehydrogenase) was used for normalization and the relative expression level was calculated using the $2^{-\Delta\Delta Ct}$ method [20]. RT-qPCR analyses of the TissueScan Cancer Survey cDNA Array 96-I (Origene, CSRT301) plates were performed by following the manufacturer's protocol and using the primer pairs of DEK1-forward and Reverse-2; DEK2-forward-2 and Reverse-3; ACTB-Forward and ACTB-reverse, provided with the kit. ACTB (Beta-Actin) expression was used for normalization and the relative expression level was calculated using the -ΔCt method. All primer sequences were shown in Table 1.

## Western blot analysis

Cells were lysed with RIPA buffer (150 mM NaCl, 1% NP40, 0.5% Sodium Deoxycholate, 0.1% SDS, 50 mM Tris-HCl pH = 8 and 1X protease/phosphatase inhibitor cocktail (Halt™ Protease and Phosphatase Inhibitor Cocktail, EDTA-free (100X), Thermo Fisher Scientific, 78441), sonicated twice for 15 seconds (sec) and incubated on ice for 30 minutes (min). Protein samples (30μg) were separated using 10% Tris-HCL (Biorad, 4569033) or %12 Bis-Tris polyacrylamide gel and transferred to nitrocellulose membranes (Biorad, 162011). The membranes were incubated overnight at 4°C with the primary antibodies diluted at 1:1000 (Anti-Flag M2 (Sigma-Aldrich, F1804), anti-DYKDDDDK Tag (9A3) (Cell Signaling Technology (CST), Danvers, MA, USA, 8146), anti-Myc (71D10) (CST, 2278), GAPDH (D16H11) (CST, 5174)) and with the HRP-conjugated secondary antibodies diluted at 3:10000 (anti-mouse IgG (CST, 7076S) or anti-rabbit IgG (CST, 7074)) for 1 hr at room temperature (RT). Membranes were visualized with enhanced chemiluminescence reagent (Advansta San Jose, CA, USA, K-12045-D50) using ChemiDoc XRS+ (Biorad, 721BR04545) system and analyzed with Image Lab. 4.0.1 software.

## Co-Immunoprecipitation (Co-IP) assay

Alternatively tagged isoforms (within MIG-DEK1 or MIG-DEK2 plasmids) were co-transfected into 293T cells using ProFection Mammalian Transfection System (Promega, E1200). Co-IP was performed using the Universal Magnetic Co-IP Kit (Active Motif Inc., Carlsbad, CA, USA, 54002) following the manufacturer's instruction and using the 2μg of anti-Flag M2 (Sigma-Aldrich, F1804) antibody.

## Cell proliferation assay

Cells were seeded in triplicate ($5 \times 10^3$ cells/200 μl per well of a 96-well plate). At the end of each corresponding day, 20 μl of WST-1 reagent (Roche, 05015944001) was added and incubated for 4 hr. Absorbance values were measured at 450 nm and 690 nm wavelength using a Varioscan microplate reader (Thermo Fischer Scientific, VL0000D0). The values of blank and the 690 nm wavelength were subtracted from the values obtained at 450 nm for each corresponding well. The final values for all time points (days) were normalized by dividing by the value of day-0 (time point at cell seeding).

## Immunofluorescence staining

Cells were fixed in 4% paraformaldehyde (PFA) for 20 min at RT, washed twice with PBS, and permeabilized with 0.5% Triton X-100 (Sigma-Aldrich, T8787) prepared in PBS for 15 min at RT. After blocking in 2% bovine serum albumin (BSA) (Sigma- Aldrich, A2153) for 1 hr at RT, primary antibody was applied for overnight at 4°C (γH2AX 1:1000 (CST, 2577S); anti-Flag M2 1:500 (Sigma-Aldrich, F1804); anti-Myc 1:1000 (9B11) (CST, 2276S), anti-Myc 1:500 (71D10) (CST, 2278S), NPM1 1:400 (Invitrogen, Thermo Fisher Scientific, 32–5200). Secondary antibodies diluted as 1:500 and incubated for 1 hr at RT (AlexaFluor-555 (CST, 4409S or 4413S), AlexaFluor-647 (CST, 4410S or 4414S)). For double immunostaining, after the first primary antibody incubation cells were re-fixed with 4% PFA for 5 min, washed and blocked with 2% BSA for 30 min at RT, and then the other primary antibody was applied. Cells were mounted with Vectashield media that contains DAPI (Vector Lab., Burlingame, CA, USA, H-1200). Images were captured with Zeiss LSM 880 Confocal Microscope (Oberkochen, Germany).

## DNA damage induction

Cells were treated with 50 nM doxorubicin (Sigma-Aldrich, D1515) (dissolved in water) or vehicle (media) for 24 hr (for recovery, cells were washed twice with PBS and cultured for an additional 24 hr with fresh growth medium (without doxorubicin)). Double immunofluorescence staining using the antibodies against γH2AX and Flag or γH2AX and Myc was performed. Cells that were double-positive for γH2AX and the corresponding epitope were used in the analyses. Images were captured by using Zeiss LSM 880 Confocal Microscope (Oberkochen, Germany) and analyzed using ImageJ software.

Endogenous expression of *DEK*s in parental HS-27A cells that were treated with 50 nM doxorubicin or vehicle for indicated times was determined by RT-qPCR using DEK1-forward and Reverse-2; DEK2-forward-2 and Reverse-3; GAPDH-forward and GAPDH-reverse primer sets.

## Doxorubicin IC50 analysis

Cells were seeded in triplicate as $5x10^3$ cells in 100μl per well of a 96-well plate. 48 hr after the treatment with various concentrations of doxorubicin, 10 μl of WST-1 (Roche, 05015944001) was added to the wells and incubated for an additional 4 hr. Absorbance (450 nm and 690 nm) was measured using a Varioscan microplate reader (Thermo Fischer Scientific, VL0000D0). The half-maximal inhibitory concentration (IC50) values were calculated by analyzing the obtained absorbance values (as described in the proliferation assay) using the GraphPad Prism 8 software (San Diego, CA, USA).

## Cell cycle analysis

Cells ($2\times10^5$ /well) were seeded in a 6 well plate and incubated for 24 hr before staining. The cells were washed with cold PBS, fixed with 70% cold ethanol for 30 min at 4˚C, and washed with PBS. After treating with 20 μg/ml RNase A (Thermo Fisher Scientific, EN0531) prepared in PBS-T (PBS including 0.1% Triton-X (Sigma-Aldrich, T8787) for 5 min at RT, cells were stained with 20 μg/ml PI (Thermo Fisher Scientific, P3566) prepared in PBS-T for 10 min at RT and analyzed by using BD Accuri™ C6 Plus Flow Cytometer (BD Bioscience). The DNA content of the cells was determined by using ModFit LT 5.0 software (Verity Software House, Topsham, ME, USA).

## Analysis of apoptosis

Cells ($2.7x10^5$) were seeded in a 6 cm culture plate and the next day was treated with 50 nM of doxorubicin or vehicle for 24 hours. Then the cells were washed, and fresh complete media (without doxorubicin) was added for recovery. The cells were then collected at time-0 (24 hr after the doxorubicin treatment), 72 hr (recovery), and 96 hr (recovery) and resuspended at a density of $1\times10^6$ cells/ml in 1X binding buffer (10mM HEPES/NaOH, pH 7.4; 140mM NaCl; 2.5mM CaCl2). 100 μL of cells were stained with 5μL Annexin V-APC (BD, 550474) and 5μL 7AAD (Biolegend, San Diego, California, U.S., 420403,) for 15 min at RT. Then, 300 μL of 1X binding buffer was added and apoptosis was analyzed using BD Accuri™ C6 Plus Flow Cytometer (BD Bioscience).

Part of the same samples was centrifuged at 4000 rpm for 2 min and pellets were lysed with Triton X-100 lysis buffer and analyzed by Western blot using anti-PARP1 (1:2000, CST) and anti-rabbit secondary antibodies.

## Bioinformatics analysis

The transcript data of primary tumors and solid normal tissues were obtained using UCSC Xena (https://xenabrowser.net) (https://doi.org/10.1038/s41587-020-0546-8), which house The Cancer Genome Atlas (TCGA) data (https://www.cancer.gov/tcga), while 1019 cancer cell lines were extracted from the Broad Institute Cancer Cell Line Encyclopedia (CCLE) using the depmap portal (https://depmap.org) (CCLE: 10.1038/nature11003). UCSC Xena uses the GENCODE v23 for transcript annotation hence the corresponding transcript IDs for the two DEK isoforms were matched with those from the newer GENCODE v39 based on their CCDS tags. Accordingly the transcript ID for the full-length DEK1 for GENCODE v39, i.e., ENST00000652689.1, corresponded to ENST00000397239.7 in GENCODE v23. For DEK2, the transcript IDs from GENCODE v39 and v23 were the same (ENST00000244776.11). CCLE database also uses an archived version of Ensembl (http://grch37.ensembl.org/) for the transcript IDs of DEK1 and DEK2, i.e., ENST00000397239.3 and ENST00000244776.7, respectively. These transcript data were logarithmically transformed before use as log2(RSEM TPM +1) and log2(RSEM+1), respectively for TCGA and CCLE.

The data obtained from ten cancer types in TCGA corresponded to the tissue types analyzed by RT-qPCR in the present study. These TCGA cancer datasets were: BRCA (Breast invasive carcinoma), COAD (Colon adenocarcinoma), KICH (Kidney chromophobe), KIRC (Kidney renal clear cell carcinoma), KIRP (Kidney renal papillary cell carcinoma), LIHC (Liver hepatocellular carcinoma), LUAD (Lung adenocarcinoma), LUSC (Lung squamous cell carcinoma), PRAD (Prostate adenocarcinoma), THCA (Thyroid carcinoma). The comparison between tumor and normal expression values for *DEK1* and *DEK2* from TCGA was performed, separately, using The Mann-Whitney U Test. Pearson's correlation coefficients for *DEK1* (ENST00000397239.3) and *DEK2* (ENST00000244776.7) transcripts were associated with P-values using stat_cor() function. All P-values were based on two-sided statistical analyses, and a P-value less than 0.05 was considered to indicate a statistically significant difference. Scatter plots for transcripts corresponding to *DEK1* and *DEK2* were plotted using the ggscatter () function from the "ggpubr" package (Alboukadel Kassambara (2020). ggpubr: 'ggplot2' Based Publication Ready Plots. R package version 0.4.0. https://CRAN.R-project.org/package= ggpubr). Beeswarm plots for the TCGA and cancer tissue array data were generated using the geom_quasirandom() function from the "ggbeeswarm" package (Clarke E, Sherrill-Mix S (2017). _ggbeeswarm: Categorical Scatter (Violin Point) Plots_. R package version 0.6.0, https://CRAN.R-project.org/package=ggbeeswarm). All computations and plots were done in R version 4.1.3 using CCLE, TCGA, and tissue array data. In addition, we used an online tool called SmulTCan (http://konulabapps.bilkent.edu.tr:3838/SmulTCan/) [21] that calculated a multivariable Hazard Ratio (HR) for each of the two isoforms of DEK, based on the mRNA expression levels of ENST00000652689.1 and ENST00000397239.7 from UCSC Xena database for available cancers in TCGA for disease specific survival (DSS).

## Statistical analysis

Statistical analyses were performed using the GraphPad Prism 8 software. Two-way ANOVA, Paired t-test or Mann-Whitney U tests were used to compare datasets.

## Results

### DEK2 locates in the nucleolus

We compared the expression levels of endogenous *DEK1* and *DEK2* in the available cell lines HS-27A and HS-5 (bone marrow stromal), 293T and HepG2 (epithelial), U937, RPMI-8226,

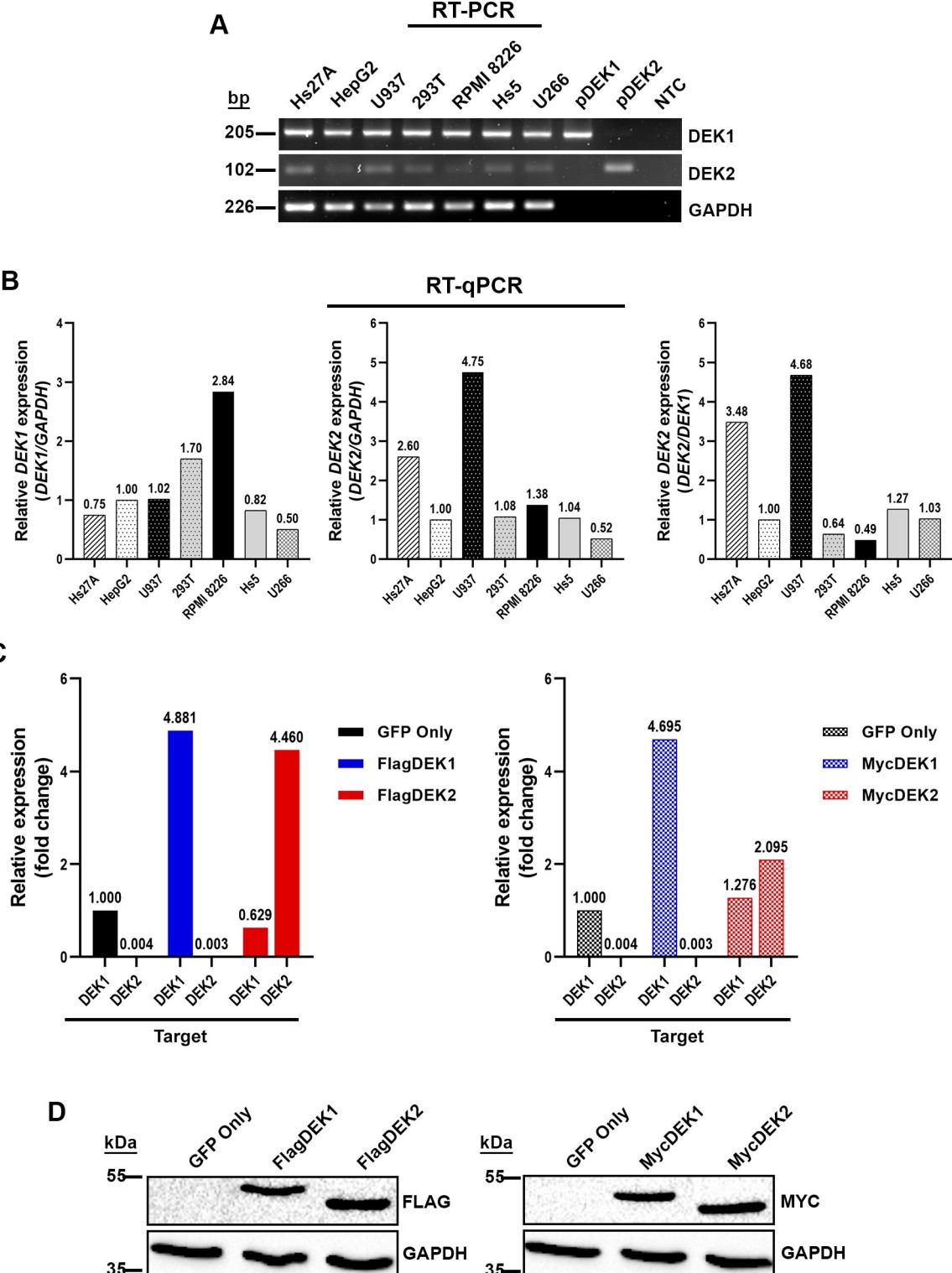

**Fig 1. Expression level of DEK isoforms.** (A) RT-PCR results show endogenous mRNA expression levels of each *DEK* isoform in human cell lines. The same amount of RNA and cDNA was analyzed for each sample by using primers specific to either *DEK1* or *DEK2* mRNA. *GAPDH* analysis was performed for the normalization. (B) RT-qPCR analyses of the same cell lines that were shown in the upper panel. HepG2 cells were used as a calibrator in these analyses. *DEK1* and *DEK2* expression that was normalized with *GAPDH* (left and middle panel) or *DEK2* expression that was normalized with *DEK1* (right panel) were shown. (C) The expression level of *DEK1* and *DEK2* in HS-27A cell lines that overexpress Flag (left panel) or Myc (right panel) tagged *DEK* isoforms was analyzed by RT-qPCR. The

expression level of each isoform was normalized with *GAPDH* expression. Fold change was calculated by using the expression level of the *DEK1* isoform in the control cells (GFP-only) as a calibrator. "Target" indicates the primer pairs that were specific to each isoform, recognizing both ectopic and endogenous mRNAs. (D) Western blot analyses show ectopic expression of Flag (left panel) or Myc (right panel) epitope-tagged DEK1 and DEK2 in HS-27A cells. Anti-GAPDH antibody was used to show equal loading.

and U266 (hematopoietic) using RT-PCR or RT-qPCR. We found that the *DEK2* expression was low in these cell lines, and both HS-27A and U937 cells had the highest level of *DEK2* (Fig 1A and 1B). Therefore, we stably overexpressed the N-terminally Flag or Myc-tagged DEK1 or DEK2 in HS-27A cells (HS-27A-DEK1 and HS-27A-DEK2) (Fig 1C and 1D).

Immunofluorescence staining with anti-Flag or anti-Myc antibodies showed that DEK2 diffusely locates in the nucleus and creates dense nuclear foci (observed in over 80% of the cells) whereas the DEK1 locates in the nucleus (observed in over 90% of the cells) (Fig 2A and 2B) (Table 2). Double-immunofluorescence staining indicated that DEK2 co-localize with the Nucleophosmin-1 (NPM1), a marker of the nucleolus (Fig 2B). Similarly, transient transfection of the HeLa cells with plasmids encoding the DEK1GFP or DEK2GFP fusion proteins showed co-localization with NPM1 in the DEK2-cells (Fig 2C). Co-immunoprecipitation assays indicated that although DEK1 self-interacts strongly, DEK2 doesn't form a detectable level of homodimers and only a small amount of DEK1 and DEK2 molecules form heterodimers (Fig 2D and 2E).

## DEK2 overexpressing cells are more sensitive to doxorubicin

Next, we examined the proliferation of HS-27A cells stably overexpressing either FlagDEK2 or FlagDEK1, which were sorted by fluorescence-activated cell sorting (FACS). Under steady-state growth conditions, both DEK1-cells and DEK2-cells showed a moderate but statistically significant (at day 5) decrease in proliferation, compared to the control cells (Fig 3A). Although the proliferation of DEK2-cells was slower compared to the DEK1-cells, the difference was not statistically significant (Fig 3A). Cell cycle analysis revealed a slight accumulation in the G2/M phase in both DEK1 and DEK2-cells (Fig 3B). When we challenged the cells with doxorubicin (which induces DNA double-strand breaks; DSBs), we found that DEK1-cells were significantly less sensitive to doxorubicin treatment (Fig 3C), which was consistent with the finding showing that downregulation of DEK1 sensitizes cancer cells to genotoxic agents including doxorubicin [7, 14]. Interestingly, DEK2-cells were more sensitive to doxorubicin when compared to the DEK1-cells (Fig 3C). These data suggest that the change in the expression balance between the two isoforms has different outcomes, and a shift that increases only the DEK2 expression results in higher sensitivity to doxorubicin when compared to the DEK1 overexpression.

## DEK2 overexpression does not compensate for the loss of DEK1

Since the HS-27A-DEK2 cells were more prone to doxorubicin-induced cell death, we analyzed if the response to DNA damage differs between the DEK1 and DEK2 overexpressing cells by assessing the γH2AX signal (phosphorylated histone H2AX, which labels the damaged DNA areas on the chromatin) using immunofluorescence staining. Since both retroviral-MIG (does not allow antibiotic selection) and lentiviral-shRNA constructs contain GFP, we first generated DEK1 or DEK2 overexpressing cells. Then, after sorting GFP+ cells by FACS, we suppressed DEK expression in these HS-27A-DEK2 or HS-27A-DEK1 cells by DEK-specific lentiviral sh-RNA (shDEK) [19] and selected the cells with puromycin to generate HS-27A-shDEK+DEK2 or HS-27A-shDEK+DEK1 cells. We couldn't achieve DEK1 overexpression in

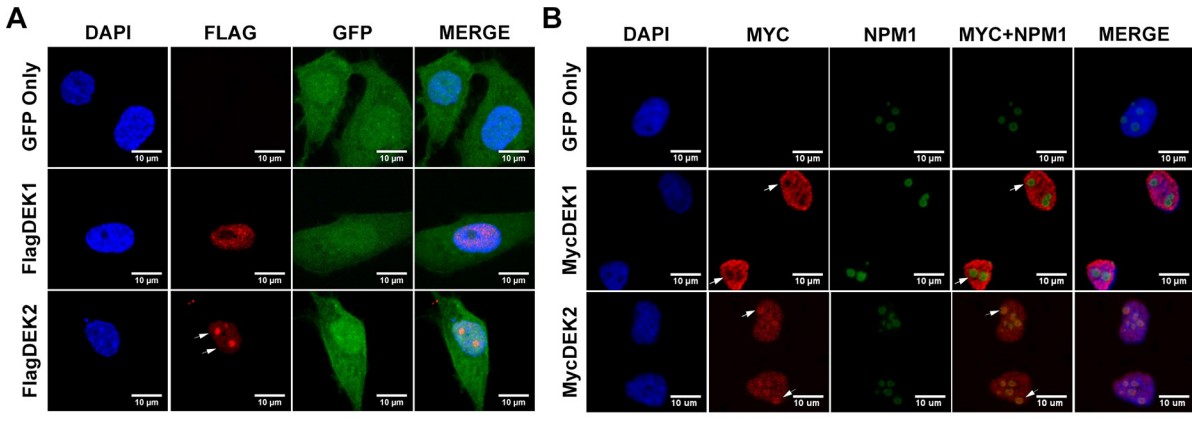

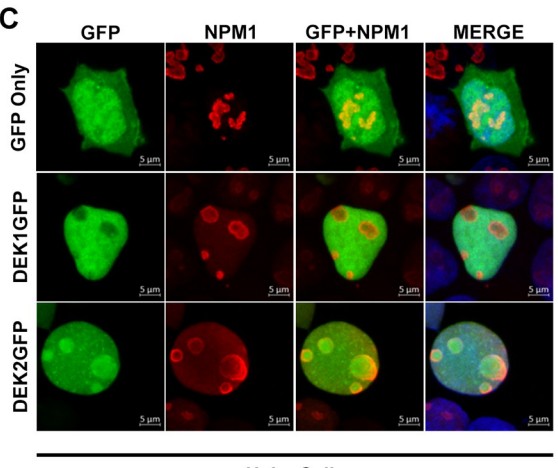

**Fig 2. Cellular localization of DEK isoforms.** (A, B) HS-27A cells stably expressing Flag-tagged (A) or Myc-tagged (B) DEK isoforms were labeled with anti-Flag or anti-Myc antibodies (red) and anti-NPM1 antibody (green in panel B) using immunofluorescence staining. GFP (green) shows transduction with the retrovirus carrying *DEK1* or *DEK2* cDNA. DAPI (blue) labels DNA in the nucleus. (C) Immunofluorescence analysis shows the expression and cellular location of ectopic DEK1GFP or DEK2GFP fusion proteins (green) in transiently transfected HeLa cells. Nucleolar marker (labeled with anti-NPM1 antibody (red)) co-localizes with DEK2GFP fusion protein. (D) Co-immunoprecipitation (Co-IP) assays using anti-Flag and anti-Myc antibodies showed that despite the interaction between the large number of FlagDEK1 and MycDEK1 molecules, only a small number of FlagDEK2 and MycDEK1 molecules interact. (E) Co-IP assay in the cells that transiently co-expressed FlagDEK2 and MycDEK2 indicated that DEK2 doesn't form homodimers.

**Table 2. Subcellular location of ectopic DEK isoforms.**

| HS-27A cells | Total cell counts | Nuclear location (cell counts / %) | Nuclear aggregates (cell counts / %) |
|---|---|---|---|
| FlagDEK1 | 216 | 216 / 100 | 0 |
| FlagDEK2 | 183 | 38 / 20.7 | 145 / 79.2 |
| MycDEK1 | 135 | 121/ 89.6 | 14 / 10.3 |
| MycDEK2 | 147 | 2 / 1.3 | 145 / 98.6 |

the presence of shDEK (S2 Fig) since the DEK-specific sh-RNA showed more efficient knock-down of *DEK1*, but we were able to overexpress DEK2, and we generated HS-27A-shDEK +DEK2 cells that have about 70% reduction in the endogenous *DEK1* expression (Fig 4A and 4B). After the 24 hr of doxorubicin treatment, analysis of anti-Flag and anti-γH2AX double-positive cells showed an increase in the intensity of γH2AX signal in the treated cells compared to the vehicle group (untreated) (Fig 4C and 4D). Surprisingly, the intensity of the γH2AX signal in HS-27A-shDEK+DEK2 cells remained significantly higher than both control (sh-negative; GFP-only) and HS-27A-shDEK cells even after removal of the drug for 24 hr (Fig 4D and 4E). These data suggest that ectopic DEK2 couldn't compensate for the loss of DEK1 and gives rise to prolonged DNA damage signal in doxorubicin treated HS-27A-shDEK+DEK2 cells.

Similar experiments with DEK1 or DEK2 overexpressing cells (HS-27A-DEK1 and HS-27A-DEK2) indicated significantly more DNA damage after 24 hr of doxorubicin treatment in both group of cells compared to the control (HS-27A-GFP; GFP-only), and the level of damage in DEK1-cells was also higher than the DEK2-cells, although statistically insignificant (Fig 5A and 5B). Despite the higher DNA damage, DEK1-cells had a better repair capacity compared to both control and the DEK2-cells, although the difference was significant only between the DEK1 and DEK2 cells (Fig 5C). DEK2-cells were more prone to cell death and exhibited a moderate increase of Annexin-V/7-AAD and cleaved p89-PARP (Poly (ADP-ribose) polymerase) staining when the recovery period after doxorubicin was extended to 96 hours (Fig 5D and 5E). In summary, DEK2-cells were more sensitive to doxorubicin than the DEK1-cells, possibly due to lower DNA repair capacity, which favors cell death. Finally, analysis of the effect of doxorubicin on the expression level of endogenous *DEK1* and *DEK2* in parental HS-27A cells by RT-qPCR revealed that the *DEK2* mRNA level was increased by 3.7-fold at 24 hr of treatment and decreased upon 24 hr of recovery. The same pattern was true for *DEK1* but only with a moderate (about 1.5-fold) upregulation (Fig 5F).

## *DEK1* and *DEK2* expression levels are correlated and frequently modulated in different cancers

We showed that there was a significantly positive correlation between *DEK1* and *DEK2* expression across a large number of CCLE cell lines from different tissues (Fig 6A). *DEK1* and *DEK2* expression levels measured by RT-qPCR in the tissue array containing both tumor and normal samples from eight cancers suggested that *DEK1* exhibited higher expression than *DEK2* (Fig 6B) as in the case of cell lines (Fig 6A). We observed significantly higher expression of *DEK1* in tumors than normal samples only in colon cancer (Fig 6B). Since the numbers of normal tissues in the tissue array were relatively low, we tested the difference between tumor and normal expression levels also in the relevant TCGA datasets (Fig 6C). Our findings indicated that tumors of colon, liver and lung exhibited higher expression of *DEK1* as compared to normal tissues. On the other hand, tumors of kidney papillary and chromophobe, prostate and thyroid had lower expression of both *DEK1* and *DEK2* while breast tumors exhibited significantly

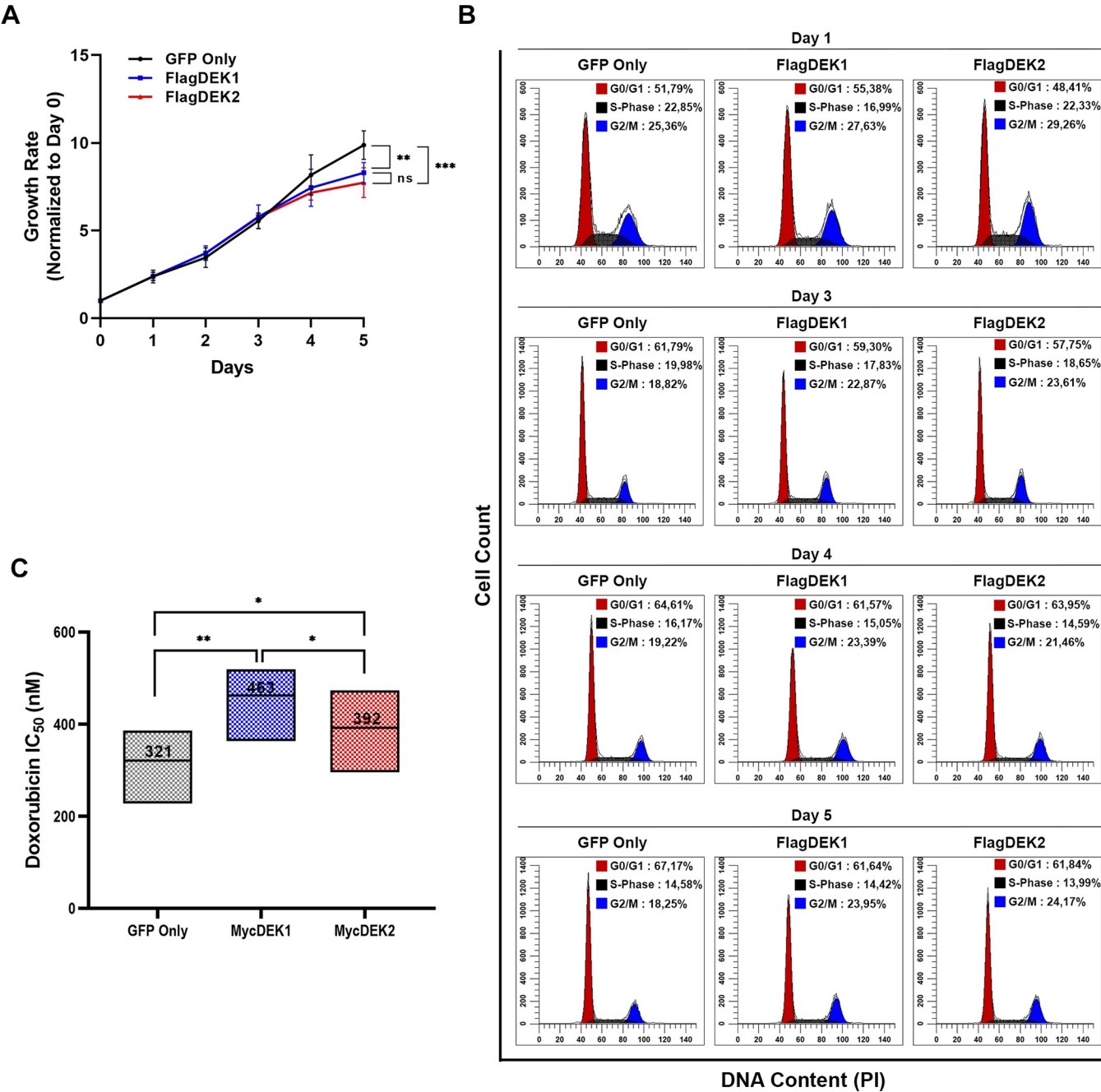

**Fig 3. Stable overexpression of DEK isoforms slows the proliferation and affects the response to doxorubicin.** (A) The growth curve of HS-27A cells stably expressing the FlagDEK1 or FlagDEK2 was generated by following the cell proliferation for 5 consecutive days using WST-1 assay. Cells transduced with an empty vector (GFP-only) were used as a control. The growth rate on each day was calculated by dividing each day's absorbance by the absorbance of Day-0. The graphic shows mean values (± SEM) obtained from three independent experiments, each performed as triplicates (Two-way ANOVA Tukey's multiple comparisons test: **P = 0.005; ***P<0.001, ns: not significant). (B) Parallel to growth curve analysis, the cell cycle was also analyzed using the same cells on days 1, 3, 4, and 5. Graphics reveal the percentage of cells at each stage of the cycle on the corresponding days. (C) Doxorubicin dose-response (IC50 values) of HS-27A-MycDEK1 or MycDEK2 cells were analyzed by using WST-1 assay. The graphic indicates the mean of three independent experiments (± SD, Paired t-test: *P = 0.0420 (MycDEK1 vs. MycDEK2); P = 0.0132 (GFP-Only vs. MycDEK2), **P = 0.0027).

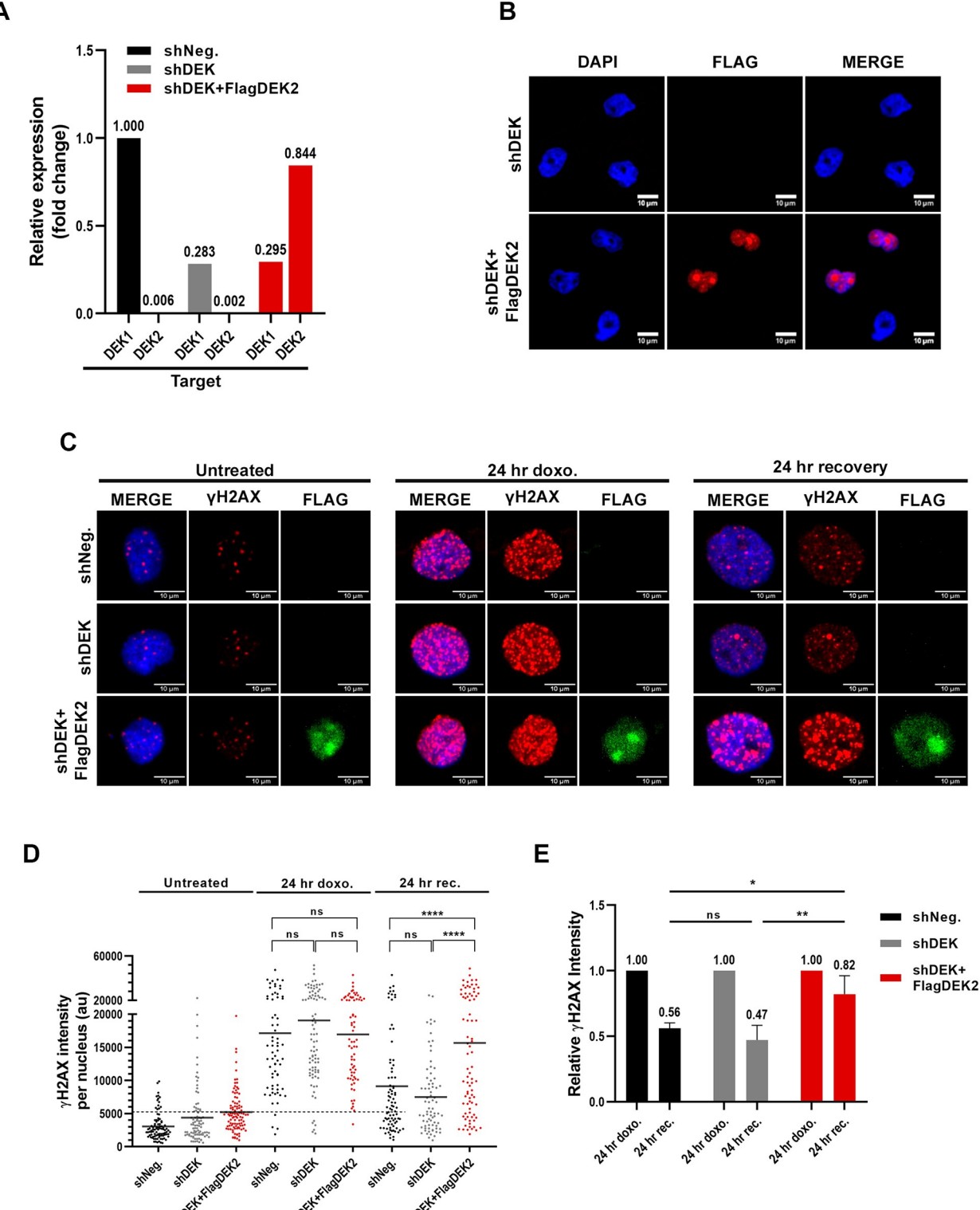

**Fig 4. DEK2 does not compensate lack of DEK1.** (A) RT-qPCR analysis showing DEK knockdown efficiency. Primer sets specific to *DEK1* or *DEK2* (X-axis, target) were used to determine the expression level of each isoform. Relative expression was calculated using the *DEK1* expression level in the control HS-27A cells (GFP only) as a calibrator. (B) Confocal pictures of immunofluorescence staining with anti-Flag antibody showing ectopic FlagDEK2 expression in the shDEK+FlagDEK2 cells (mid-low panel, red). DAPI labels the nucleus (blue) (40X, with oil). (C) Representative confocal pictures of double immunocytochemistry analysis showing staining with anti-γH2AX (red) and anti-Flag (green)

antibodies in the vehicle (untreated) or doxorubicin (24 hr doxo.) treated cells. Cells recovered for 24 hr after doxorubicin treatment depicted as "24 hr recovery". DAPI staining (blue) labels the nucleus (40X, with oil). (D) Representative graphic showing anti-γH2AX staining intensity per nucleus of each cell (Y-axis of the graph; au: arbitrary units) that was determined by using Image-J software. Each dot in the graph represents a cell and between 50 and 100 cells were analyzed per each corresponding cell line (X-axis) (bar indicates mean value. Mann Whitney test: ****P<0.0001, ns: not significant). (E) Reduction in the γH2AX signal intensity after the 24 hr of recovery from the doxorubicin treatment was determined by dividing the mean intensity of the recovery cells (24 hr rec.) by the mean intensity of the doxorubicin-treated (24 hr doxo.) cells in each group. The graph indicates the mean of two independent experiments (error bar indicates ± SD. Two-way ANOVA Tukey's multiple comparisons test: *P = 0.0325, **P = 0.0087).

lower levels of only *DEK2* based on luminal-A and normal-like subtypes when compared to the normal samples (Fig 6D). Moreover, multivariable Cox Regression based HR analyses performed with SmulTCan web server [21] (screens 33 TCGA-PANCAN datasets containing melanoma and liver hepatocellular carcinoma) revealed that higher *DEK1* expression levels were significantly and more often associated with poor prognosis of cancer patients (8 out of 9 cancer types), than *DEK2* expression (S1 Table). However, higher *DEK2* expression was an indicator of better prognosis for KIRC and BRCA while was hazardous for pancreatic adenocarcinoma (PAAD) (S2 Table).

## Discussion

Here we provide pieces of evidence that distinguish DEK1 and DEK2 isoforms by their subcellular location and function. We showed that the DEK2 locates in the nucleus and nucleolus, doesn't form a homodimer but interacts with DEK1, and it couldn't functionally replace DEK1 in the DEK-knockdown cells that were damaged by doxorubicin treatment.

The full-length *DEK (DEK1)* is expressed ubiquitously and easily detected in many cells, whereas *DEK2* expression is much lower (Figs 1 and 6). About 4 to 6 million DEK1 molecules exist per nucleus, and 2 to 3 of these molecules associate with nucleosomes [22, 23]. In many cancers, overexpression of DEK1 is associated with increased proliferation and poor prognosis [5–8]. Here, we have found that *DEK1* and *DEK2* expressions are correlated with each other in cancer cell lines and tumors yet tissue specifically, and the expression level of *DEK1* is higher than that of *DEK2*. Interestingly, there was a tendency towards lower *DEK2* expression in general, and in breast invasive tumors there was a significant downregulation of *DEK2* in luminal-A and normal-like tumors (Fig 6D) suggesting that *DEK2* can be modulated differently than *DEK1* in specific cancers. In support of this, we show a significant increase only in *DEK1* levels in colorectal cancers by RT-qPCR (Fig 6B) and using the TCGA-COAD dataset (Fig 6C). TCGA-liver and lung carcinoma datasets also show higher *DEK1* levels in tumors as supported by previous studies [24–26]. However, our findings suggest that *DEK1* but not *DEK2* is overexpressed in these three types of epithelial cancers. *DEK1* and *DEK2* expression levels being lower in the tumors than normal tissues in kidney, prostate and thyroid carcinoma is another novel finding not yet reported in the literature. Moreover, we have found that *DEK2* is not expressed in a considerable number of tumors in TCGA dataset that might warrant further investigation.

To investigate whether DEK1 and DEK2 are functionally distinguishable, we overexpressed epitope-tagged DEK2 in the HS-27A cell line in which either endogenous expression of DEKs was protected or suppressed. In agreement with published data [23, 27], we showed that ectopic DEK1 locates mainly in the nucleus. On the contrary, we found that DEK2 locates both in the nucleus and nucleolus (Table 2 and Fig 2B and 2C). Posttranslational modifications of DEK1, such as phosphorylation, acetylation, and poly-ADP-ribosylation reduce DEK's

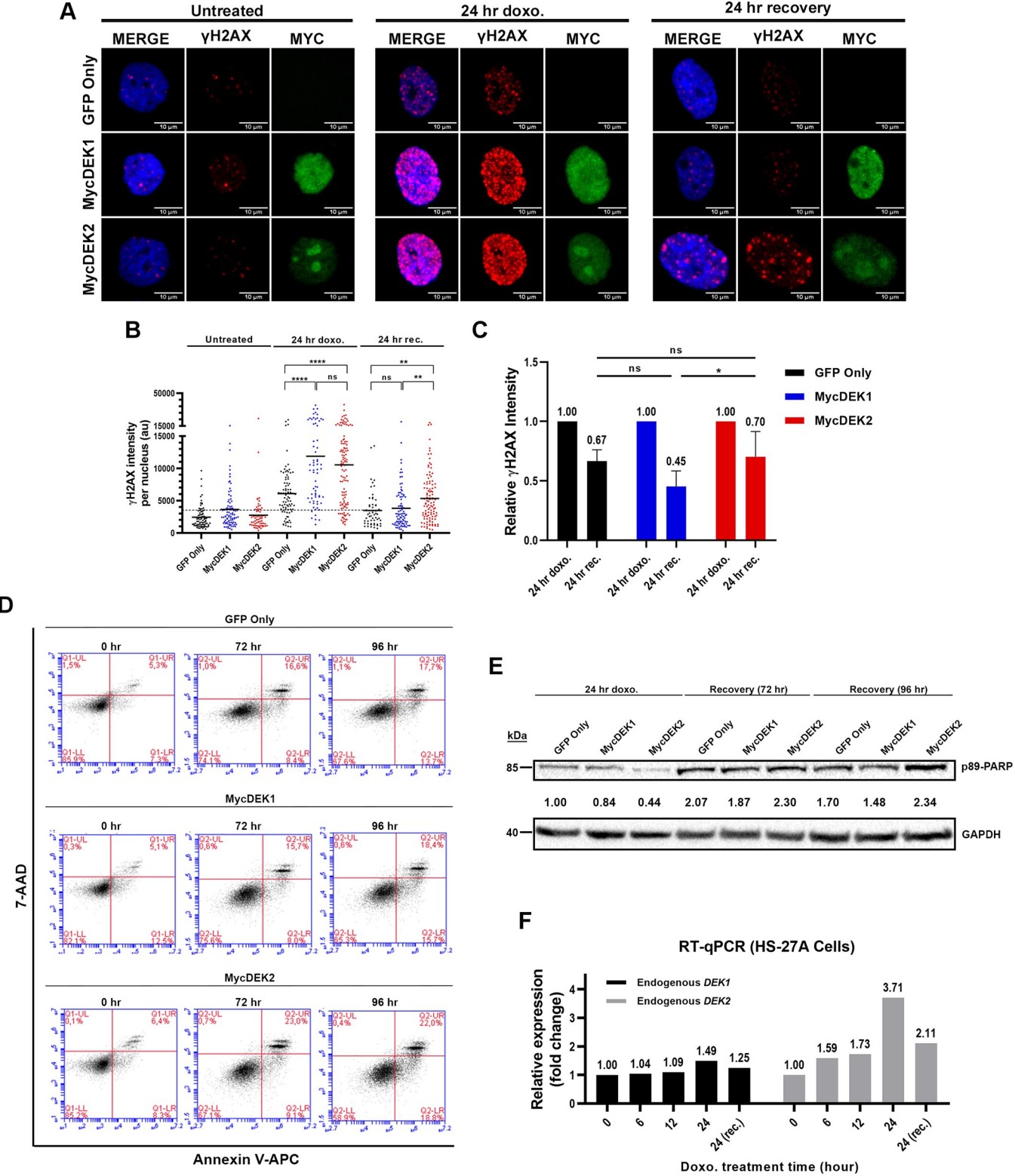

**Fig 5. Effect of DEK1 or DEK2 overexpression on DNA repair efficiency.** (A) MycDEK1 (anti-Myc antibody, green in middle panels), MycDEK2 (anti-Myc antibody, green in lower panels), and γH2AX expression (anti-γH2AX antibody, red) were determined by immunocytochemistry and representative pictures captured by confocal microscope were shown in the vehicle (untreated), doxorubicin (24 hr doxo.) and recovery (24 hr recovery) groups (40X, with oil). DAPI (blue) labels the nucleus. (B) A graphic of one representative experiment shows the anti-γH2AX staining intensity per nucleus (Y-axis) in each group that was

determined by using Image-J software. Each dot represents a cell and between 50 and 100 cells were analyzed in each group (bar indicates mean value. Mann Whitney test: **P = 0.0010 (GFP-only vs. MycDEK2), **P = 0.0011 (MycDEK1 vs. MycDEK2), ****P<0.0001, ns: not significant). (C) Reduction in the γH2AX signal intensity after the 24 hr of recovery from the doxorubicin treatment was determined by dividing the mean intensity of the recovery cells (24 hr rec.) by the mean intensity of the doxorubicin-treated (24 hr doxo.) cells in each group. The graph indicates the mean of three independent experiments (error bar indicates ± SD. Two-way ANOVA Tukey's multiple comparisons test: *P = 0.0384). (D) Annexin-V analysis after the 24 hr doxorubicin (50 nM) treatment (0 hr) and following 72 hr or 96 hr recovery of control (GFP only), MycDEK1 or MycDEK2-cells. (E) Western blot analysis of the same cells shown in D exhibited an increased level of cleaved p89-PARP in the MycDEK2 cells. Anti-GAPDH was used to show equal protein loading. Quantification of band intensity was calculated using Image-J software (PARP/GAPDH). Fold change was determined by using GFP-only (24 hr doxo) sample as a calibrator and indicated between the Western blot images of PARP and GAPDH. (F) RT-qPCR analysis showing the endogenous *DEK1* and *DEK2* expression levels in HS-27A cells that were treated with 50nM doxorubicin for the indicated time points (X-axis). Twenty-four hr of recovery (24 (rec)) was also applied after the 24 hr of doxorubicin treatment. Vehicle-treated cells for each time point were used as a calibrator in the RT-qPCR analyses.

affinity to chromatin and facilitate interaction with its partners [13, 14, 28]. In addition to other binding partners [29], DEK also self-interacts [30] and the C-terminal amino acids between 270–350 are required for dimerization. This region contains casein kinase 2 (CK2) phosphorylation sites, which promote self-interaction [30]. Our co-immunoprecipitation assays revealed that DEK1 self-interacts strongly but also forms a weak DEK1-DEK2 complex (Fig 2D). On the other hand, DEK2 doesn't self-interact (Fig 2E), and whether the lack of amino acids 49–82 also affects its binding to DNA/chromatin and other partners of DEK1 is yet to be determined.

The role of DEK on DSBs-repair was previously investigated upon short-term DNA damage induction (1hr) and recovery (4 to 8 hr). It was shown that repression of DEK1 expression reduces DNA repair as judged by persistent γH2AX signal in the knockdown cells [14, 31]. Here we showed that doxorubicin-induced DSBs were repaired similarly in the control and shDEK cells when the recovery time was extended to 24 hr (Fig 4E). The longer period and DEK-knockdown efficiency could yield a better recovery from doxorubicin-induced damage in our setting.

A comparison of endogenous *DEK2* and *DEK1* mRNA levels in doxorubicin-treated cells has shown a temporary increase, more strongly in *DEK2*, followed by a decline after the removal of the drug suggests a possible role in the damage-alarming system (Fig 5F). Given that ectopic *DEK2 or DEK1* expression moderately slows the proliferation (Fig 3A) and induces accumulation at the G2/M phase (Fig 3B), upregulation of endogenous *DEK* isoforms by doxorubicin might contribute to the induction of cell cycle arrest, which is required for the repair of damaged DNA. Supporting this idea, a constitutively high amount of DEK2 results in a long-lasting DNA damage signal (Figs 4D and 5B; 24 hr rec). Although human DEK1 rescue the DNA repair defect in *Dek*-knockout mouse embryonic fibroblasts [31], our data indicate that DEK2 couldn't functionally compensate the loss of DEK1 (in shDEK cells), and DEK isoforms might have a non-overlapping contribution to DNA repair. Supportively, DEK1 overexpressing cells are less sensitive to doxorubicin-induced cell death (Fig 3C) and they repair DNA damage more efficiently (Fig 5B and 5C), which was consistent with previous findings [7, 14, 32].

In summary, our findings suggest that expression of *DEK2* might be modulated differently than *DEK1* in a tissue specific manner and both isoforms might have different functions. The structural change might drive DEK2 to the nucleolus and affect its binding to proteins that interact with DEK1, leading to functional differences that we observed. Future studies aiming to identify their binding partners and generation of *DEK2* knock-in or *DEK2*-transgenic mouse models would allow a better understanding of DEK-isoforms' role in different cellular processes.

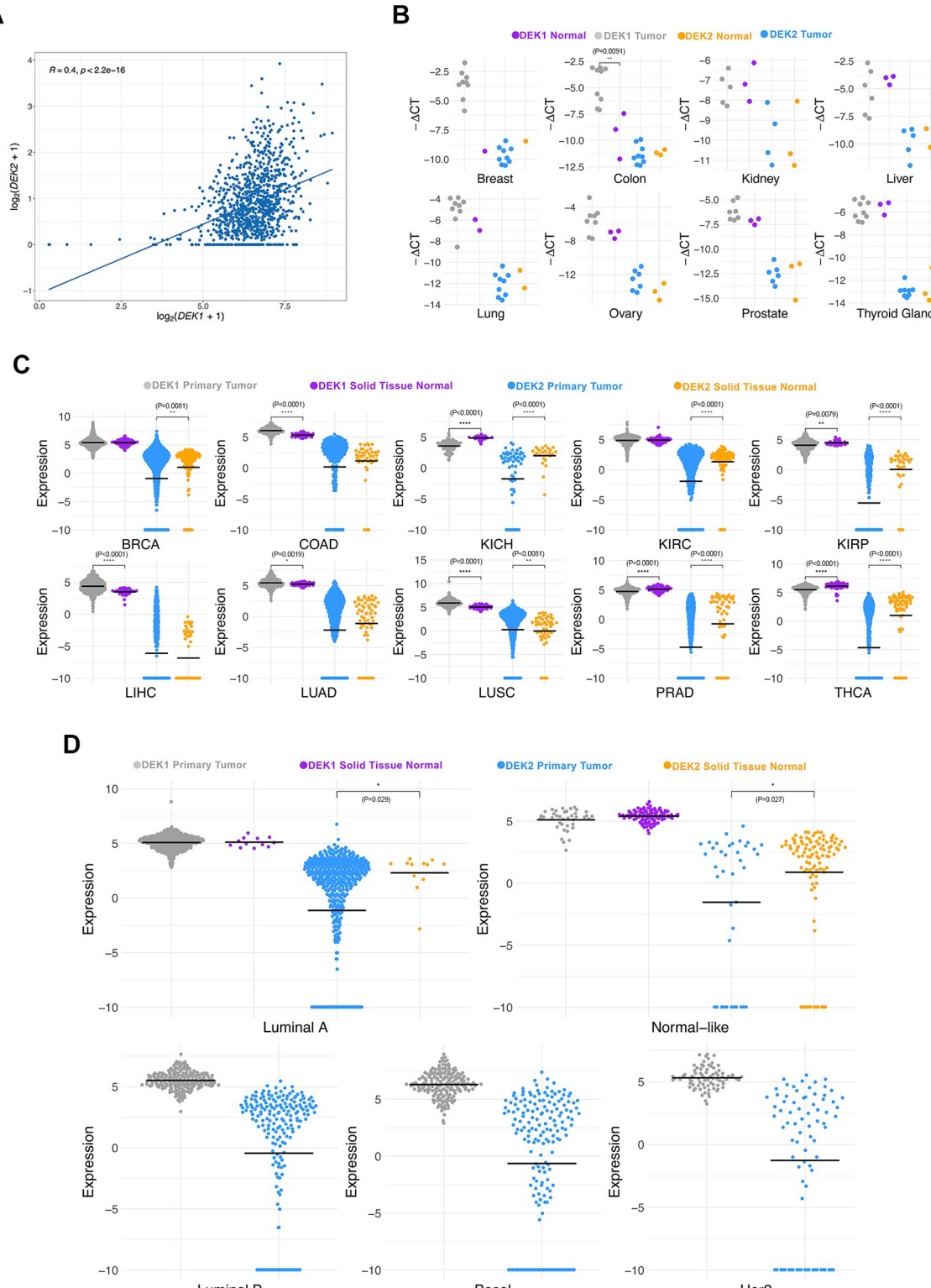

**Fig 6. The mRNA expression profiles of *DEK1* and *DEK2*.** A) CCLE dataset; B) Tissue array panel; and C, D) TCGA datasets. *DEK1* and *DEK2* Ct values were normalized to *ACTB* (tissue array) and reported as -ΔCt while CCLE and TCGA dataset transcript values were transformed as log2(RSEM+1) and log2(RSEM TPM+1), respectively. The Mann-Whitney U Test was used for statistical analysis. Asterisks represent the statistical significance.

## Supporting information

**S1 Fig. FACS analysis indicates percentage of GFP⁺ cells after the FACS-sort.**
(TIF)

**S2 Fig. FlagDEK1 expression level in shDEK+FlagDEK1 cells.**
(TIF)

**S1 Table. Hazard ratios of DEK1 for disease specific survival.**
(TIF)

**S2 Table. Hazard ratios of DEK2 for disease specific survival.**
(TIF)

**S1 Raw images.**
(PDF)

## Acknowledgments

The results shown here are in part based upon data generated by the TCGA Research Network: https://www.cancer.gov/tcga. The authors thank Prof. Dr. Uygar Halis Tazebay for helpful discussions, Dr. Ömer Güllülü, Doğukan Metiner and Mehmet Soner Türküner for technical assistance.

## Author Contributions

**Conceptualization:** Emrah Özçelik, Ayten Kandilci.

**Funding acquisition:** Emrah Özçelik, Ayten Kandilci.

**Investigation:** Emrah Özçelik, Ahmet Kalaycı, Büşra Çelik, Açelya Avcı, Hasan Akyol, İrfan Baki Kılıç, Türkan Güzel, Metin Çetin, Merve Tuzlakoğlu Öztürk, Zihni Onur Çalışkaner, Melike Tombaz, Dilan Yoleri, Özlen Konu.

**Project administration:** Ayten Kandilci.

**Resources:** Ayten Kandilci.

**Supervision:** Özlen Konu, Ayten Kandilci.

**Writing – original draft:** Emrah Özçelik, Özlen Konu, Ayten Kandilci.

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
