## [Decision Letter · Decision Letter 0]

10 Aug 2022

PONE-D-22-18362Doxorubicin induces prolonged DNA damage signal in cells overexpressing DEK isoform-2.PLOS ONE

Dear Dr. Kandilci,

Thank you for submitting your manuscript to PLOS ONE. After careful consideration, we feel that it has merit but does not fully meet PLOS ONE’s publication criteria as it currently stands. Therefore, we invite you to submit a revised version of the manuscript that addresses the points raised during the review process.

We look forward to receiving your revised manuscript.

Kind regards,

Sekyu Choi

Academic Editor

PLOS ONE

Journal Requirements:

"This work was supported by the Scientific and Technological Research Council of Turkey (TÜBİTAK) (grant numbers 118Z765 and 216Z006). The authors thank Prof. Dr. Uygar Halis Tazebay for helpful discussions, Dr. Ömer Güllülü, Doğukan Metiner and Mehmet Soner Türküner for technical assistance."

We note that you have provided funding information that is not currently declared in your Funding Statement. However, funding information should not appear in the Acknowledgments section or other areas of your manuscript. We will only publish funding information present in the Funding Statement section of the online submission form. Please remove any funding-related text from the manuscript and let us know how you would like to update your Funding Statement. Currently, your Funding Statement reads as follows: 

"Prof. Dr. Ayten Kandilci

Grant number: 118Z765

Funding: Scientific and Technological Research Council of Turkey (TÜBİTAK)

https://www.tubitak.gov.tr/

Mr. Emrah Özçelik

Grant number: 216Z006

Funding: Scientific and Technological Research Council of Turkey (TÜBİTAK)

https://www.tubitak.gov.tr/

5. We note you have included a table to which you do not refer in the text of your manuscript. Please ensure that you refer to Table 2 in your text; if accepted, production will need this reference to link the reader to the Table.

Reviewers' comments:

Reviewer's Responses to Questions

**Comments to the Author**

1. Is the manuscript technically sound, and do the data support the conclusions?

Reviewer #1: Yes

Reviewer #2: Yes

2. Has the statistical analysis been performed appropriately and rigorously? 

Reviewer #1: Yes

Reviewer #2: I Don't Know

3. Have the authors made all data underlying the findings in their manuscript fully available?

Reviewer #1: Yes

Reviewer #2: Yes

4. Is the manuscript presented in an intelligible fashion and written in standard English?

Reviewer #1: Yes

Reviewer #2: Yes

5. Review Comments to the Author

Reviewer #1: Özçelik et al. stably overexpressed DEK2 in the human bone marrow stromal cell line, and found that DEK2, unlike DEK1, is present in the nucleus and nucleolus, and that doxorubicin treatment causes persistent γH2AX signaling, which cannot functionally compensate for the loss of DEK1. Cells overexpressing DEK2 were found to be more sensitive to doxorubicin than DEK1 cells, and the expression of DEK1 and DEK2 in cell lines and primary tumors showed tissue specificity. In additon, only DEK2 is downregulated in a subset of breast cancers, suggesting that DEK2 may be differentially regulated from DEK1 in certain cancers. The authors concluded that the expression patterns and subcellular locations of the two DEK isoforms are different, suggesting that there is no overlap in function. This paper is well-written, however, some points are not clarified.

1. If DEK1 would be silenced, how does the DEK2 expression change?

2. In normal tissues, usually DEK1 expresses in the proliferative zone of the epithelium. The DEK2 expresses the same region? For example, how about colon (glandular) and esophagus (squamous cell) epitheliums?

3. It is interesting that only DEK2 is downregulated in a subset of breast cancers but not DEK1. Localization of two DEK proteins would be shown by double immunofluorescent staining in cell lines and/or human tissues.

Reviewer #2: This article explores the biology of a quite enigmatic variant of DEK: DEK2. The authors are investigating the similarities and differences between these 2 isoforms, and whether their expression levels could be of interest in cancer studies. Overall, this study is well conducted and well controlled. However, there are a few issues that the authors should consider revising.

The cloning section in the mat&met is confusing: was DEK2 cDNA obtained from cells or by site-directed mutagenesis using DEK1 cDNA? This point should be clarified. Moreover, the bioinformatics analysis section in the mat&met is not sufficiently detailed and requires additional information such as which exact datasets have been used and their respective URL. Also, the identifier ENST00000397239.3 for DEK1 is incorrect and should be amended.

On line 258, Table 1 should be changed for Table 2.

With figure 2B-D, the authors are willing to show the co-localisation between DEK2 and NPM1 in the nucleolus. However, from my point of view, the lack of NPM1 staining in figure 2C makes this panel irrelevant. The authors should consider adding NPM1 staining or removing this panel.

The authors are claiming that “DEK1-cells were resistant to doxorubicin-induced cell death” (lines 279-280). I do agree that the data showed that these cells are significantly less sensitive to doxorubicin, nevertheless they are still responsive to this treatment. Thus, I think this point should be amended.

To the review question “2. Has the statistical analysis been performed appropriately and rigorously?” I answered “I don’t know” because I think a mistake was made in figure 4E. Indeed, all statistics indicate non-significant, but looking at the other panels and the text, I am convinced there are some significance there, isn’t it? Moreover, the authors are claiming on lines 313-314 that “the level of damage in DEK1-cells was higher than the DEK2-cells” but statistics are missing.

The authors only commented on the better recovery of DEK-1 cells compared to DEK2-cells. However, DEK1-cells present a significant reduction of gH2AX intensity compared to control-cells too. This point should be discussed.

Since p89-PARP expression presents only slight variations in figure 5E, it would be interesting to see a quantification.

Finally, DEK oncogene has already been studied in melanoma and hepatocellular carcinoma, for example. It would then be interesting to consider the potential involvement of DEK2 in the discussion.

6. PLOS authors have the option to publish the peer review history of their article (what does this mean?). If published, this will include your full peer review and any attached files.

Reviewer #1: No

Reviewer #2: No

---

## [Author Response · Author response to Decision Letter 0]

2 Sep 2022

We thank the reviewers for all suggestions/questions and comments, which helped us to improve our manuscript. We have responded at our best to all of the points raised by reviewers in the revised manuscript and /or in the response letter (shown in blue).

---

## [Editor Report · Decision Letter 1]

19 Sep 2022

Doxorubicin induces prolonged DNA damage signal in cells overexpressing DEK isoform-2.

PONE-D-22-18362R1

Dear Dr. Kandilci,

We’re pleased to inform you that your manuscript has been judged scientifically suitable for publication and will be formally accepted for publication once it meets all outstanding technical requirements.

Kind regards,

Sekyu Choi

Academic Editor

PLOS ONE

Additional Editor Comments (optional):

I am satisfied the author's responses to all reviewers' comments.

They changed the manuscript based on reviewers's comments.
---

## [Editor Report · Acceptance letter]

22 Sep 2022

PONE-D-22-18362R1 

Doxorubicin induces prolonged DNA damage signal in cells overexpressing
DEK isoform-2. 

Dear Dr. Kandilci:

I'm pleased to inform you that your manuscript has been deemed suitable for publication in PLOS ONE. Congratulations! Your manuscript is now with our production department. 

Kind regards, 

on behalf of

Dr. Sekyu Choi 

Academic Editor

PLOS ONE